# Recurrent neural networks learn robust representations by dynamically balancing compression and expansion

**Matthew Farrell**
Department of Applied Mathematics
University of Washington
msf9@uw.edu

**Stefano Recanatesi**
Department of Physiology and Biophysics
University of Washington
stefanor@uw.edu

**Guillaume Lajoie**
Department of Mathematics and Statistics
Université de Montréal
lajoie@dms.umontreal.ca

**Eric Shea-Brown**
Department of Applied Mathematics
University of Washington
etsb@uw.edu

## Abstract

Recordings of neural circuits in the brain reveal extraordinary dynamical richness and high variability. At the same time, dimensionality reduction techniques generally uncover low-dimensional structures underlying these dynamics. What determines the dimensionality of activity in neural circuits? What is the functional role of dimensionality in behavior and task learning? In this work we address these questions using recurrent neural network (RNN) models . We find that, depending on the dynamics of the initial network, RNNs learn to increase and reduce dimensionality in a way that matches task demands. These findings shed light on fundamental dynamical mechanisms by which neural networks solve tasks with robust representations that generalize to new cases.

Recordings of neural circuits in the brain reveal extraordinary dynamical richness and high variability. At the same time, dimensionality reduction techniques generally uncover low-dimensional structures underlying these dynamics. What determines the dimensionality of activity in neural circuits? What is the functional role of dimensionality in behavior and task learning? In this work we address these questions using recurrent neural network (RNN) models. We find that, depending on the dynamics of the initial network, RNNs learn to increase and reduce dimensionality in a way that matches task demands. These findings shed light on fundamental dynamical mechanisms by which neural networks solve tasks with robust representations that generalize to new cases.

## 1   Introduction

Dynamics shape computation in brain circuits. Due to the limitations in our ability to record every neuron in a circuit, it can be difficult to characterize these dynamics through direct observation alone. Bridging between machine learning and neuroscience, artificial recurrent neural networks (RNNs) are powerful tools for investigating dynamical representations in controlled settings, and enable tests of theoretical hypotheses that can be leveraged to formulate experimental predictions (reviewed in [2]). Thinking of artificial networks as dynamical brain circuits is likewise a useful way of understanding their power and flexibility. Since RNNs give rise to well-defined dynamical systems, the *neural representation* of the recurrent units is governed by the system's dynamical response to inputs. In this work we task a network with classifying inputs into one of two classes (binary classification). We

33rd Conference on Neural Information Processing Systems (NeurIPS 2019), Vancouver, Canada.

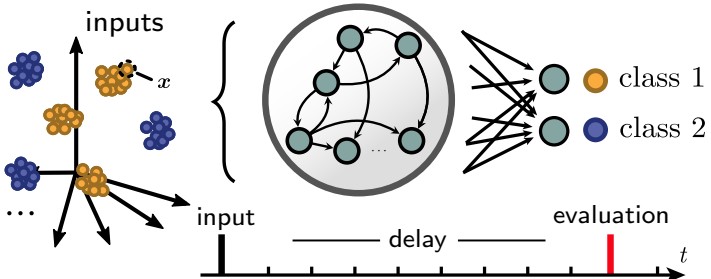

Figure 1: Task and model schematic.

treat each input as an impulse delivered at an initial time $t_0$, and allow the RNN a delay period to process this input before querying the network to output the class label (Fig. 1).

To reveal the essential dynamical elements at play in high-dimensional systems such as RNNs, dimensionality reduction is routinely employed [4]. These approaches reveal a surprising fact: rather than scaling with the number of neurons in the circuit, dynamics are often effectively constrained to regions whose dimensionality seems to be intimately linked to the complexity of the function, or behavior, that the neural circuit fulfills or produces [11] . This link between task and representation dimension is especially intriguing in light of fundamental ideas in learning theory. On one hand, high-dimensional representations can subserve complex and general computations that nonlinearly combine many features of inputs [6, 12]. On the other, low-dimensional representations that preserve only essential features needed for specific tasks can allow learning based on fewer parameters and examples, and hence with better generalization [6, 15].

Here we ask how an RNN balances reducing and increasing dimensionality of input data, and link this behavior to network dynamics. We find that the answer can depend on initialization; in particular, networks that are initially more *chaotic* have a tendency to expand the dimensionality of low-dimensional inputs. Frequently encountered in network models of brain function, dynamical chaos (whereby tiny changes in internal states are amplified by unstable, but deterministic, dynamics) provides a parsimonious explanation for both repeatable structure as well as internally generated variability seen in highly recurrent brain networks such as cortical circuits [14, 7]. While chaos-driven dimensionality expansion with fixed recurrent weights has previously been explored [8], the attributes of this phenomenon as recurrent weights evolve through training are less understood.

## 2   Results

We consider a standard RNN architecture with hidden unit dynamics $h_t$ and outputs $o_t$ given by

$$h_t = \tanh(Rh_{t-1} + x_t + b_1)$$
$$o_t = Wh_t + b_2$$

where $\tanh$ is applied element-wise. Here $R$ and $W$ are the recurrent and output weights, respectively, $b_1$ and $b_2$ are biases, and $x_t$ is the input. Simulations are run with $n = 200$ hidden units. The weights $R$ are initialized as a random perturbation of the identity to ensure smooth trajectories: $R^0 = (1 - \varepsilon)I + \varepsilon J$ where $\varepsilon = .01$ and $I$ is the identity matrix. The matrix $J$ has normally distributed entries that scale in magnitude with a coupling strength parameter $g$: $J_{ij} \sim \mathcal{N}(0, g^2/n)$. The coupling strength $g$ governs the degree to which the network is chaotic, with higher values leading to more chaotic dynamics. The outputs are initialized at random: $W_{ij}^0 \sim \mathcal{N}(0, .3^2/n)$.

Inputs are drawn from Gaussian distributed clusters lying within a $d$-dimensional random subspace of the $n$-dimensional neural activity space. The means of these clusters are distributed uniformly at random within a bounded region of this subspace. Each cluster is assigned one of two class labels, as illustrated in Fig. 1. While this schematic shows six clusters for clarity, in our simulations we use 60 clusters with half being assigned label 1 and half label 2. Recurrent and output weights, as well as biases, are adjusted via stochastic gradient descent with momentum (RMSProp) to minimize a cross-entropy loss function, and classification performance is evaluated on held-out inputs. The learning rate is initially taken to be large enough to cause significant changes in the loss, and is modified by a reduce on plateau strategy through training.

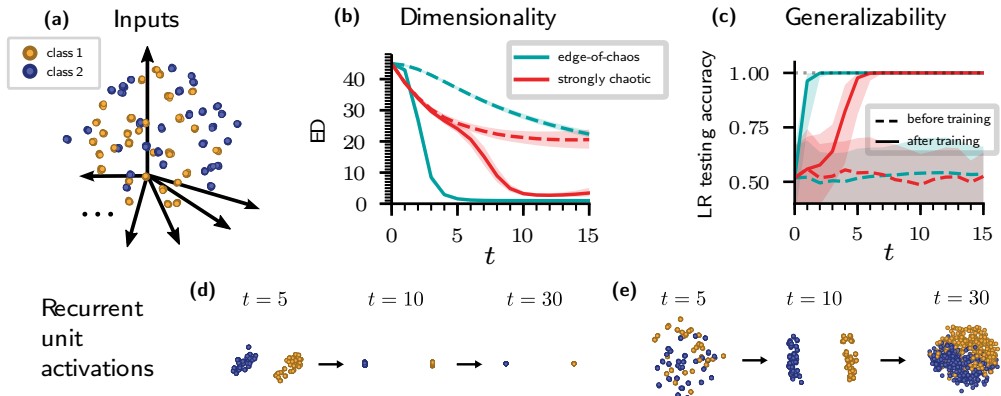

Figure 2: Comparison of edge-of-chaos (blue-green) and strongly chaotic (red) networks classifying high-dimensional inputs. Input color (yellow-orange or purple) denotes true class label. Shaded regions indicate 75% probability mass of gamma distribution fits over 30 network and input realizations, with lines indicating medians. (a) Visual schematic of inputs. (b) Effective dimensionality (ED) of the network representation through time $t$. Dashed and solid lines denote before and after training the network, respectively. (c) Mean testing accuracy of an LR classifier on network response to held-out input clusters. Dashed lines and legend as in (b). (d) Activations of recurrent units responding to inputs, plotted as "snapshots" in time in PC space, for the edge-of-chaos network. (e) Same as (d) but for the strongly chaotic network.

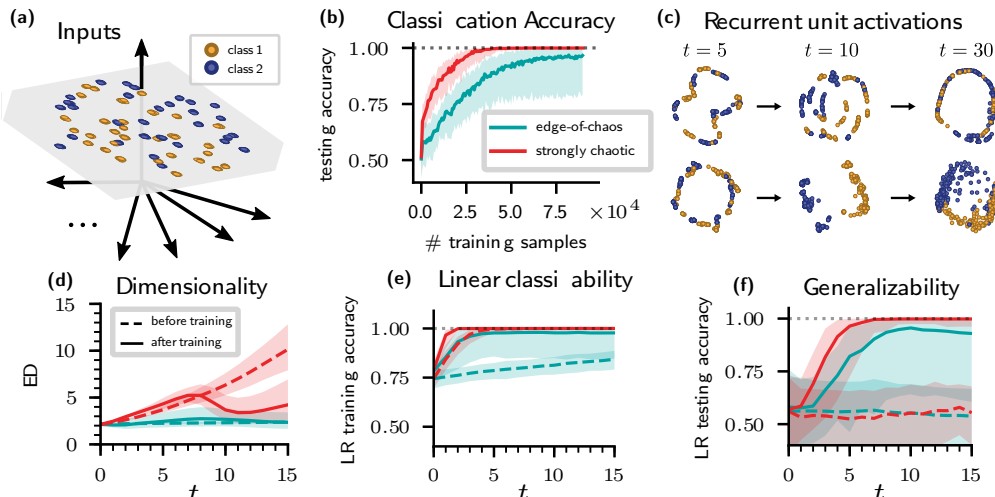

Figure 3: Comparison of edge-of-chaos (blue-green) and strongly chaotic (red) networks classifying low-dimensional inputs. (a) Inputs lie on a two-dimensional plane. (b) Testing accuracy. (c) Top: as in Fig. 2d. Bottom: as in Fig. 2e (d) As in Fig. 2b (e) Mean training accuracy of a logistic regression (LR) linear classifier trained on the recurrent unit activations at each timepoint $t$. (f) As in Fig. 2c

**RNNs learn to compress high-dimensional inputs**

We start by considering the classification of high-dimensional inputs $d = n$ that allow a linear class separation boundary (Fig. 2a). We compare a network initialized near the edge of chaos (EOC) to one that is initially strongly chaotic (SC), corresponding to coupling strength parameters $g = 20$ and $g = 250$, respectively. Both networks achieve perfect testing accuracy after training (not shown). Fig. 2d shows the top two principal components (PCs) of the network responses to 600 input samples at snapshots in time. The EOC network forms two stable fixed point attractors, one for each class. The SC network separates the classes into two chaotic attractors that begin to mix back

together after the evaluation time $t_{\text{eval}} = 10$. This compression phenomenon is partly captured by the *effective dimensionality* (ED) of the representation [9]. The equation for the ED of a set of points is $\text{ED} = 1/\left(\sum_i \tilde{\lambda}_i^2\right)$ where $\tilde{\lambda}_i = \lambda_i / \sum_j \lambda_j$ are the normalized eigenvalues of the covariance matrix of the points. ED can be roughly thought of as the number of PCs needed to capture most of the variance of the points. In Fig. 2b the ED of the representation at times $t$ is plotted over $t$. The EDs of the trained networks are approximately equal to that of the input at time $t = 0$, since the initial states only differ from the inputs by one application of the nonlinearity. The dimensionality drops with increasing $t$ and is highly compressed at the evaluation time $t_{\text{eval}} = 10$. This drop results both from increasing distances between different classes as well as decreasing distances within classes. Similar phenomena have recently been reported in deep neural networks [3]. These phenomena appear to be fairly general, occuring in networks trained via a mean squared error loss as well as networks where the hyperbolic tangent nonlinearity is replaced with a linear activation function (not shown). While we find these results robust to moderate changes in hyperparameters, it should be noted that the degree of dimensionality reduction as well as the chaoticity of the networks after training depends somewhat on the chosen learning rate. In particular, higher learning rates tends to dampen chaos more and increase dimensionality compression (not shown).

We next study the coding properties of these representations. To capture how close the representations are to being linearly separable, we train a logistic regression (LR) linear classifier on the representations at each timepoint. In this case, the LR classifier achieves perfect training accuracy on the representations of the SC and EOC networks for all time up to at least $t = 15$ (not shown), both before and after the networks are trained. This confirms that linear separability is preserved by the network dynamics. The interesting properties of the compressed representation can be seen in Fig. 2c, which measures generalization by first training an LR classifier on the network response to inputs drawn from a fixed 80% of the input clusters, and then measuring the accuracy of this classifier on the network response to samples drawn from the remaining 20% of the clusters. The dashed lines indicate that the representations of the untrained networks do not generalize well to held-out clusters, even while they allow for linear separation boundaries. In contrast, after training, the network representations become increasingly generalizable through time $t$, eventually allowing for perfect classification accuracy on held-out clusters.

**Strongly chaotic RNNs learn to expand and then compress low-dimensional inputs**

We next consider inputs embedded in a two-dimensional space, $d = 2$ (Fig. 3a). In this case, boundaries must be highly curved and nonlinear to separate the classes. Fig. 3b illustrates a surprising fact: the SC network learns to classify near-perfectly, while the EOC network is not as successful. PC plots (Fig. 3c) indicate that the SC network is better at separating the classes. This seems to result from the increasing dimensionality through time $t$ exhibited by the SC network in Fig. 3d, much more so than for the EOC network. After training, the SC network first increases dimensionality up to time $t = 7$, and then decreases up to time $t = 11$. This allows it to achieve linear separability of the two classes (Fig. 3e) as well as good generalization (Fig. 3f). The non-monotonic behavior of the ED depends on the learning rate, since too high (say, doubling) of a learning rate can quench chaos while too low (say, halving) won't cause a dimensionality reduction after the expansion (not shown). These phenomena also appear in networks trained via a mean squared error loss, although we find that the EOC network can learn to expand dimensionality and solve the task unless the task is made more challenging (the SC network solves the task as before). When the task is made more challenging by adding more clusters, there is a regime where the EOC network does not learn to expand dimensionality and fails to solve the task with high accuracy while the SC network does solve the task with high accuracy (not shown).

## 3   Conclusions and Future Directions

We find that in tasks where inputs are linearly separable by class label, RNNs generically reduce the dimension of their inputs over the delay period, in the process forming a representation that lends itself to good generalization. Next, we find that in harder tasks where inputs are low-dimensional and class separation boundaries are highly nonlinear, only networks with sufficiently chaotic initializations are successful. We explain this by showing that chaos-driven dimensionality expansion results in representations with linear separation boundaries. Taken together, we find evidence that RNNs

learn representations that have the minimal dimensionality needed to support relatively simple class separation boundaries, provided that the initialization is sufficiently chaotic. These findings invite further exploration of learning strategies through the lens of modulating dimensionality and suggest functional roles for variability found in brain circuits.

While effective dimensionality was chosen in this study because it is able to capture the distribution of disjoint manifolds and related coding properties, in general it is of interest to study nonlinear measures of dimensionality (i.e. *intrinsic* dimension). Recent work has explored this direction in the context of deep feedforward neural networks [10, 1], but connection between nonlinear dimension and RNN dynamics have still not been explored as far as we are aware. In addition, it is of interest to track the dimension of individual input or class manifolds, as is done in [3] for deep feedforward networks. It is also of interest to see if the phenomena of dimension compression and expansion can be captured by mathematical analysis. See [13, 5] for work in the direction of demonstrating how compression can be driven by stochastic gradient descent. While this study suggests roles for dimensionality modulation and (chaotic) variability in biological neural circuits, it would be interesting to look for this explicitly in experiments. Finally, it is of interest to see if these phenomena extend to (recurrent) network models that achieve state-of-the-art performance, and to see if the principles explored here can be used to improve the functioning of such networks.

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
