# OpenReview forum: "Recurrent neural networks learn robust representations by dynamically balancing compression and expansion"
_NeurIPS.cc/2019/Workshop/Neuro_AI — Real Neurons & Hidden Units @ NeurIPS 2019 Poster_

### Official Review · AnonReviewer2 · 2019-09-24
**recurrent neural nets can expand or compress the input space through their dynamics**

**Clarity:** 4

**Comment:**

The important idea suggested in this paper is that RNNs can show both behaviours, compressive and expansive, depending on their initial chaotic state. The smart choice of toy examples in the paper helped drawing an intuitive picture of the hypothesis and the RNNs dynamics, and also supporting the main hypothesis.
Since this paper attributes a functional importance to chaotic behaviour of neuronal circuitries in the brain, to clarify the importance of this theory for neuroscience, it is crucial to explain how the chaotic state of different circuitries in the brain can be potentially modulated in different tasks or contexts. Do different brain circuitries exhibit different levels of chaotic dynamics? Also, since expansion and compression of representation space are differently preferred for different tasks (as shown by the two toy examples in the paper), can a single neuronal circuitry manage to do both in different contexts (if yes, how)? These issues are not addressed in this study which could be considered as a limitation, though given the limited space of the paper, it's understandable, and can be considered for future steps.
Also, the other important question concerns the role of training/learning in the empirical observations on low- and high-dimension neuronal representations in the brain. As shown in this paper, the compressive or expansive representations have been formed after training the RNNs on the task. Therefore, one could speculate that the empirical observations on the dimensionality of neuronal activity could also be the by-product of training animals on the experimental tasks. This can be considered as another reason for probing neuronal activity not only on well-trained animals but also throughout the training process.


**Category:**

AI->Neuro

**Clarity Comment:**

The motivation behind their study is clear. The use of RNN as a model of brain dynamics is well justified.
The schematics and result figures are clear and their line of reasoning is easy to follow.


**Evaluation:**

4: Very good

**Importance:**

4: Very important

**Importance Comment:**

The low dimensional manifolds (e.g. in frontal regions) as well as high dimensionality of representations (e.g. mixed selectivities in prefrontal regions) both have been shown to exist in the brain, and based on the learning theory, both have their own computational advantages. In this paper, the authors show that recurrent neural networks (RNN), as models of brain dynamics, can form both low- and high-dimensional representation spaces depending on their initial pre-training chaotic behaviour.

**Intersection:**

3: Medium

**Intersection Comment:**

The contribution of this paper, in its current format, is mainly to neuroscience than AI.
This study brings a very important insight to the neuroscience community which became possible by a smart application of AI models to a neuroscience problem. Even though this study was motivated by a neuroscientific problem (low dimensional manifolds or high dimensional mixed selectivities in the brain), the findings of this paper can potentially lead to a better understanding of RNN behaviour for AI community as well.


**Rigor Comment:**

The main hypothesis is that the initial chaotic behaviour of RNNs matter in their post-training representation space. Through simulations, the authors provided two toy examples where RNNs were trained to solve a binary classification problem in a high dimensional input space. The RNN at the edge of chaos (EOC) can form a compressed representation space that suits the binary classification problem. Intuitively, the EOC RNN collapsed all the input space to two attractors that were easily separable. In contrast, the strongly chaotic RNN (SC) learned to expand the input space and form a linearly separable representation of input samples. The SC RNN was shown to be quite useful when the input samples were lying on a low-dimensional manifold in the input space where the two classes were not linearly separable (Figure 3). The examples support the main hypothesis that the two initial modes of chaotic behaviour leads to two different learned representations in RNNs. However, it's not yet convincing that this model underlies the observed dimensionality (low or high) of neuronal activity in the brain. Specifically, since the two expansive and compressive behaviours depend on two different initial chaotic behaviours, and since different tasks might need expanded or compressed representation spaces (as the examples in the paper show), it's hard to imagine how the chaotic state of different brain circuits could change in a controlled way (or is this even what one could suggest?).
Furthermore, no mechanistic explanation is provided by the authors for the relationship between chaotic dynamics and the dimension of learned representation space.
In short, the materials in the paper support the claim that both expanded and compressed representation spaces can be formed by RNNs, though it is not shown how chaotic dynamics are computationally linked to the dimension of RNNs’ representations, and why this hypothesis should be taken seriously by neuroscientists.


**Technical Rigor:**

3: Convincing

---

### Official Review · AnonReviewer3 · 2019-09-26
**RNN initialization interacts with task demands to influence the dimensionality of RNN dynamics**

**Clarity:** 2

**Comment:**

There are several areas where the results of the paper could be extended to make a stronger case to both communities.

A few examples:
- The difference between the two networks is present at initialization. Does this have implications for RNNs in ML, which are typically randomly initialized before training?
- The results seem to suggest that SC networks may offer performance benefits over EOC networks. Is there evidence of this?
- How do these results relate to the typical operating regime of networks in task-related circuits in the brain? E.g. how do the results suggesting chaotic networks can temporarily increase the dimensionality relate to results in [8]? The tasks in this paper are very simple, so these results may not directly generalize to more complicated settings.

**Category:**

Common question to both AI & Neuro

**Clarity Comment:**

The main result concerns differences in behavior between RNNs in edge of chaos (EOC) and strongly chaotic (SC) regimes. These terms are not defined in the paper, nor does the paper explain how networks are sampled in these two regimes.

Effective dimensionality seems like a reasonable way to measure dimensionality in this context, but it is ultimately a linear measure. Given that RNN dynamics can be highly nonlinear, it is likely to miss some nuances of network behavior. A discussion of possible limitations would help interpretation of the results.

The plot legends don't show what dashed and solid lines mean, making the plots hard to read. This is only explained in the Figure 2b caption, which I had to refer back to several times while reading the paper.

**Evaluation:**

3: Good

**Importance:**

3: Important

**Importance Comment:**

This paper characterizes RNN dynamics over the time course of response, suggesting that networks that are strongly chaotic and those at the edge of chaos behave differently. These results suggest that the operating regime of an RNN at initialization may modulate how it interacts with task dimensionality. This result is novel as far as I know. It is likely to be of interest to computational neuroscientists interested in RNNs and task dynamics and is potentially of interest to the AI community.

**Intersection:**

4: High

**Intersection Comment:**

This paper shows general results on simple RNNs, which are likely to be of interest to members of both the AI and neuro communities.

**Rigor Comment:**

The results in the paper are generally clear and experiments seem well-designed. All of the experiments are done with two networks: one initialized near the edge of chaos (EOC) and one that is initially strongly chaotic (SC). The paper doesn't explain how they obtain these two network states, so it's unclear if other properties of these two networks might be contributing to the effects observed.

It's unclear how robust the small effect of dimensionality increasing then decreasing seen in figure 3d is. Is this change reliable and significant? Is it preserved on inputs with different, low values of d or in networks with different numbers of units?

**Technical Rigor:**

3: Convincing

---

### Official Review · AnonReviewer1 · 2019-09-27
**interesting scientific question, but only preliminary, anecdotal numerical results for toy problem**

**Clarity:** 5

**Comment:**

The scientific question of this paper is fascinating and the study of generalization and classification are without flaws. However, to make the study more relevant, it would be essential to investigate if both of the two main findings (dimensionality reduction of input throughout learning, and improved classification for linearly nonseparable problems) are a general phenomenon or just a feature of the training protocol (BPTT with RMSProp), the nature of the toy problem (classification of Gaussian point clouds), and the model class (randomly initialized discrete-time vanilla RNN). To make it more relevant for the neuroscience community,  this could be investigated in a biologically more plausible model/task. Also, predictions that could be tested in experiments would be desirable.
To make it more relevant for machine learning, either rigorous results (e.g., upper/lower bounds on the dimensionality, capacity of the network, etc.) (c.f. Chung, Lee, Sompolinsky 2016, 2018) or applications to state of the art RNN problems (e.g., machine translation, NLP) using state of the art gated units would be desirable.

**Category:**

Common question to both AI & Neuro

**Clarity Comment:**

Both the scientific problem is well-explained, also the methods are very clear and the results and their relation to previous works are very understandable.


**Evaluation:**

2: Poor

**Importance:**

2: Marginally important

**Importance Comment:**

This paper addresses how the dimension of input representations is changing both across time-steps and during training and how different network initializations affect classification performance and generalization. The main numerical finding is that the dimension is reduced during training. However, inputs that are not linearly separable can be better classified by a network initialized  'strongly chaotic'. While the scientific question is important the findings seem preliminary and anecdotal.


**Intersection:**

3: Medium

**Intersection Comment:**

Questioning how, across time and during learning/training, the dimensionality of representations changes in recurrent networks is relevant both for the neuroscience and the machine learning community. However, concerning the relevance for neuroscience, the model (discrete-time, firing rates, random networks) seems far away from biological plausibility. It is not clear to what extent chaotic rate fluctuations of firing rate models can account for the variability observed experimentally. It is not clear either to what extent the finding that dimensionality of representations decreases throughout learning depends on the training protocol (backpropagation through time with RMSProp) and on the model class. It is not explained how this result relates to previous work linking the dimensionality of neural activity to the task complexity (Peiran Gao, Eric Trautmann, Byron Yu, Gopal Santhanam, Stephen Ryu, Krishna Shenoy, Surya Ganguli 2017). For the machine learning community, the scientific question: what we can learn about the computation from the dimensionality of representations; could potentially be interesting, but a more realistic problem where RNNs are having superior performance compared to feed-forward models would be more instructive. Why would one study the question in such a simple toy problem, without aiming for analytical results?

**Rigor Comment:**

The results are purely numerical and only found in on one toy problem. The dimensionality estimate is based on the covariance of the network activity across states, which does only take into account structure captured by the first two moments but seems to be standard in neuroscience (Gao, Ganguli, 2015). The training of the network is done using backpropagation through time with RMSProp minimizing a cross-entropy loss for a delayed classification task. The influences of the initialization of the weights on the classification accuracy likely depends on the training algorithm/protocol/hyperparameters, but the authors do not investigate this question. Also, the link between the initially putatively strongly chaotic vs. edge-of-chaos dynamics of the RNN and the dimensionality is not clear. Isn't chaos leading to unpredictable dynamics? Why should that help in classification unless the initial state of the network is fixed and noise-free? Isn't there possibly an issue of robustness to noise in the initial conditions? The quantification of the classification performance is to my understanding done correctly (separate training and test set). It would be desirable to check the results using nonlinear dimensionality reduction techniques (e.g. t-SNE, Isomap etc.), because PCA has many known issues. Therefore, it is not clear if this a more general phenomenon or just an interesting anecdote. Moreover, the RNN is neither biologically realistic (discrete-time, no spikes, random networks) nor state of the art in machine learning (vanilla RNN classifying Gaussian point clouds, no gated units, no test of the hypothesis on standard benchmark data sets using state of the art models).
While the dimension across input patterns is quantified similarly to previous work (Chung, Lee, Sompolinsky 2018) albeit, without analytics, a quantification of the dimension of the object manifolds (dimension within a class) is missing. It is not clear how robust the results are with respect to changes in the parameters, e.g. task complexity, dimension of object manifold. In conclusion, while the analysis seems to be done correctly, the results seem rather preliminary and based numerical anecdotal evidence rather than analytical or general mechanistic insights.

**Technical Rigor:**

2: Marginally convincing

---

### Decision · Program_Chairs · 2019-10-02

Accept (Poster)